# The Influence of Welding Heat Source Inclination on the Melted Zone Shape, Deformations and Stress State of Laser Welded T-Joints

**DOI:** 10.3390/ma14185303

**Published:** 2021-09-14

**Authors:** Zbigniew Saternus, Wiesława Piekarska, Marcin Kubiak, Tomasz Domański

**Affiliations:** 1Faculty of Mechanical Engineering and Computer Science, Czestochowa University of Technology, Dabrowskiego 69, 42-201 Czestochowa, Poland; kubiak@imipkm.pcz.pl (M.K.); domanski@imipkm.pcz.pl (T.D.); 2Faculty of Architecture, Civil Engineering and Applied Arts, University of Technology, Rolna 43, 40-555 Katowice, Poland; piekarska@imipkm.pcz.pl

**Keywords:** laser welding, numerical modeling, thermomechanical phenomena, stainless steel, deformations

## Abstract

The paper concerns the numerical analysis of the influence for three different of welding heat source inclinations on the weld pool shape and mechanical properties of the resulting joint. Numerical analysis is based on the experimental tests of single-side welding of two sheets made of X5CrNi18-10 stainless steel. The joint is made using a laser welding heat source. Experimental test was performed for one heating source inclination. As a part of the work metallographic tests are performed on which the quality of obtained joints are determined. Numerical calculations are executed in Abaqus FEA. The same geometrical model is assumed as in the experiment. Material model takes into account changing with temperature thermophysical properties of austenitic steel. Modeling of the motion of heating source is performed in additional subroutine. The welding source parameters are assumed in accordance with the welding process parameters. Numerical calculations were performed for three different inclinations of the source. One inclination is consistent with experimental studies. The performed numerical calculations allowed to determine the temperature field, shape of welding pool as well as deformations and stress state in welded joint. The obtained results are compared to results of the experiment.

## 1. Introduction

Laser beam used in the welding process is one of rapidly developed technologies in the industry. Laser welding techniques are currently widely used in the automotive, energy, chemical, petrochemical, nuclear, electronic and many other industries. Concentrated laser beam heat sources provide a narrow fusion zone (FZ) and a small heat affected zone (HAZ), which reduces the deformation of welded construction pieces in comparison to conventional welding methods. The use of laser techniques allows one to perform precise welds using extremely high welding speeds and low deformations. Therefore, the obtained joints do not require additional post processing. Laser welding is widely used in large-size constructions having innovative solutions for joining elements with complex shapes and in joint of elements made of various materials with different thermomechanical properties. The growing requirements to the quality of manufactured joints result in intensive experimental and numerical research on the laser welding process in a wide range, like the type of joints, process parameters or joined materials [1,2,3,4,5,6,7].

Large structures are created on automated production lines in production plants, where various types of welded joints are used. T-joints are a one of the most commonly used type of welded joints in large-size welded constructions. T-joints are classically performed with arc methods like fillet welding, in which it is necessary to use an additional material during the welding process. An important factor in such welding processes is the high thermal load of the element, which has an impact on the shape of the welding pool, strength properties of the joint and its weight. Low welding speeds with arc methods additionally increase production costs. The use of modern welding equipment in the production process like a laser beam has significantly increased the production capacity of T-joints [8,9,10,11,12]. Laser methods allow T-joints to be made using various welding techniques, such as classic joints with the use of double-side and single-side fillet welds, with or without additional material as well as T-joints with the use of butt-welds. The I-core T-joint is an innovative solution, in which the material is melted by laser through the flange of T-bar creating a fused “nail”, connecting elements [13,14,15]. Performed research on T-joints (numerical as well as experimental) concerns the analysis of physical phenomena in welding processes and the analysis of T-joints in complex large-size industrial constructions.

Coupled thermo-mechanical phenomena occur in a wide temperature range during the laser beam welding process, having an impact on the joint quality [16,17,18,19,20,21,22]. The strength, deformation and load-bearing capacity of welded joints are influenced by the volume and shape of FZ, which depend on the welding parameters: mainly welding speed and the amount of heat supplied to the material. It is necessary to set-up joined materials precisely in the case of laser welding without the use of additional material and with no gap between joined parts.

The formation of a gap between the connected elements may lead to the formation of a concave face of the weld, which is adverse in welded joint. The analysis of welding parameters is carried out using numerical methods, replacing expensive experimental studies in this area. Numerical simulations allow one to test various welding parameters that affect the welding process in terms of joint deformation and mechanical properties in relation to welding technique and the behavior of material of welded elements used in the process. The adaption of real process conditions in a discrete model requires the development of appropriate mathematical models and numerical solutions [23,24,25,26]. An important step of numerical modeling is the assumption of an appropriate mathematical model of laser beam power distribution heating joined parts, the direction and method of its acting on the material, and the proper selection of process parameters. Developed mathematical and numerical models should be verified by comparing obtained simulation results with experimentally measured data, such as: the shape and size of the melted zone and heat affected zone, the value of deformation and the level of residual stresses in joined elements [27,28,29]. Modeling the welding of T-joint requires consideration of the welding source inclination in computational model. The selection of appropriate parameters is ensured by experimental verification.

The work presents a numerical analysis of the influence of welding heat source inclination on the weld pool shape, stress state and welding deformation of the welded joint.

Numerical modeling of phenomena occurring in fillet welding process of a T-joint using a laser beam welding source. One of the basic parameters that has a significant impact on the quality of the joint and mechanical properties is the inclination of the welding beam. Numerical calculations were performed for three different source inclinations α = 20°, 30°and 40°. The starting point for the conducted research is a real welding experiment made using laser beam for one chosen inclination. The parameters accepted in real welding process are further used as input data to the numerical solution program. A computational model of thermomechanical phenomena of the welding process is developed with the use of the real process parameters. Numerical simulations are carried out to determine the size and shape of FZ and the deformation of the joint. Numerical simulations are made in commercial computational software Abaqus solver [30] with additional, self-made numerical subroutines. Based on the developed model, simulation calculations were carried out for the other two inclinations of the source.

## 2. The Experiment

T-joint welding processes were carried out at Research Network Łukasiewicz–Welding Institute in Gliwice. The joined elements in the welded T-joints are two flat bars of different thickness (150 mm × 30 mm × 3 mm and 150 mm × 30 mm × 1 mm), made of X5CrNi18-10 stainless steel. The flat bars are welded on one side using welding station equipped with a CO_2_ laser. A welded joint and its cross-section are presented in Figure 1 and Figure 2, respectively.

The following parameters are used in the welding process of the T-joints: laser beam source power 2100 W (read from the device during the process) and the source travel speed 1.5 m/min. The inclination angle of the welding beam is applied by 20° to the horizontal flange of the joint in order to obtain a fillet weld. The laser welding process of the T-joint was carried out with the use of argon shielding gas flow rate16 L/min.

As part of the experimental tests, the deformation of the welded joint was measured (Figure 3). The measurement of a shape of the deformed flange of T-joint was conducted in Surface Layer Laboratory in Institute of Fundamental Technological Research, Polish Academy of Science. A precise scanning profilometer Hommel-Etamic T8000 Nanoscan (Jena, Germany) was used. It enables high accuracy measurements. The maximum vertical resolution is less than 1 nm and the maximum lateral resolution equals 0.01 and 0.5 μm in the x- and y-direction, respectively. The scanned area was 27 mm in y direction and 145 mm in the x-direction and was therefore slightly smaller than that of the flange. Since we captured only the shape (a waviness, not a roughness) of the flange, we used a lower measurement resolution of 5 μm in the x-direction and 1.8 mm in the y-direction. The scanning speed was 0.5 mm/s. On the basis of the registered measurement data extract surface profiles in any direction can be obtained.

The measurements were carried out for three measurement lines in the perpendicular direction to the welding line. Figure 4a shows the measurement lines. The test results are presented in the diagram shown in Figure 4b.

## 3. Mathematical Model

The thermal analysis is based on the classical heat transfer equation, given in weighted residuals criterion method [30]:(1)$$\int_{V}\rho \frac{\partial U}{\partial t}\delta TdV+\int_{V}\frac{\partial \delta T}{\partial x_{\alpha }}\cdot \left(\lambda \frac{\partial T}{\partial x_{\alpha }}\right)dV==\int_{V}\delta T\;q_{V}dV+\int_{S}\delta T\;q_{S}dS$$
where *λ =*
*λ(T)* is thermal conductivity (W/mK), *U = U(T)* is internal energy (J/kg), *δT* is variational function, *ρ* is density (kg/m^3^), *q_v_* is volumetric heat source (W/m^3^), *q_s_* is a surface heat flux (W/m^2^).

Equation (1) is completed by boundary conditions and the initial condition *t = 0: T = T_o_*, taking into account convection and radiation:(2)$$\begin{array}{l} {T|}_{\Gamma }=\tilde{T}\;q_{Sym}=-\lambda \frac{\partial T}{\partial n}=0\\ q_{S}=-\lambda \frac{\partial T}{\partial n}=\alpha_{k}(T|{}_{\Gamma }-T_{0})+\varepsilon \sigma (T{|{}_{\Gamma }}^{4}-T_{0}^{4}) \end{array}$$
where *q*(*r*, 0) is heat flux at the top surface (*z* = 0) within the radius *r*, *T_0_* is an ambient temperature, *α_k_* is convective coefficient (W/m^2^K), *ε* is radiation, *σ* is the Stefan-Boltzman constant.

The latent heat of fusion *H_L_* = 260 × 10^3^ (J/kg) is assumed in numerical simulations in the solidus-liquidus temperatures range (*T_S_* = 1673 K; *T_L_* = 1728 K).

Assuming an appropriate mathematical model of power distribution of the heating source is an important step in developing the numerical model. Modeling of a moveable welding source is performed by implementing DFLUX subroutine (ver.1.1) into the Abaqus computational solver. A volumetric model of the source power distribution is included in the subroutine as well as welding speed and direction. In this work, Gaussian model with an energy distribution changing linearly with the material thickness is used to describe the laser beam power intensity distribution [31,32]:(3)$$Q_{v}(r,z)=\frac{P\cdot \eta }{\pi \;r_{o}^{2}h_{1}}\exp\left[1-\frac{r^{2}}{r_{o}^{2}}\right]\left(1-\frac{z}{d}\right)$$
in which:(4)$$r_{o}=r_{t}-\left(r_{t}-r_{b}\right)\cdot \frac{z}{h_{1}}$$
where *P* is a laser power (W), *r_0_* is a laser beam radius (m), *η* is efficiency of welding processes, *d* is material penetration depth (m), *r* is actual radius (m), where $r=\sqrt{x^{2}+y^{2}}$, *r_t_* and *r_b_* is beam radius, respectively for *z = 0* and *z = h_1_*, *h_1_* is a height of heat source penetration and *z* is actual depth of penetration (m) [32].

According to experimental research, the numerical model described by Equation (3) assumes a linear decrease of source energy with the depth of penetration of the material by the laser beam. The volume of laser source in the form of a truncated cone is assumed (Figure 5). Performing a fillet weld between the connected flat bars required inclination of the welding source. Therefore, numerical modelling of fillet welded also requires the inclination angle of the laser beam heat source distribution axis. It is necessary to transform the vertical axis of the beam source by the angle α adopted in the experiment. The transformation of the coordinate system is performed in DFLUX subroutine. The schema of heat source coordinate transformation is shown in Figure 6.

The transformation of welding source power distribution is carried out using transformation model:(5)$$A_{i^{\prime }}=\gamma_{i^{\prime }j}\;A_{j}\;where\;\gamma_{i^{\prime }j}=\underset{i^{\prime }}{e}\;\cdot \underset{j}{e}$$

Equations of coordinates transformations are obtained after the solution of presented transformation matrix from the basic system (*x, y, z*) to the rotated system (*x_1_, y_1_, z_1_*):(6)$$\left\{ \begin{array}{l} x=x_{o}\\ y=\;\cos\;\alpha \;\cdot y_{1}+\sin\;\alpha \;\cdot z_{1}\\ z=-\sin\alpha \;\cdot y_{1}+\cos\alpha \;\cdot z_{1} \end{array}\right.$$

In mechanical analysis, the temperature field obtained from the heat analysis is taken as the thermal load. Calculations of mechanical phenomena are carried out in the elastic-plastic range and are based on the equilibrium Equation (7). Governing equations are supplemented by constitutive relations describing the relationship between stresses and strains. Initial and boundary conditions are taken into account in the calculations [33,34,35,36]:(7)$$\begin{array}{l} \nabla \circ \dot{\sigma }\left(x_{\alpha },t\right)=0,\;\dot{\sigma }={\dot{\sigma }}^{T}\\ \dot{\sigma }=D\circ {\dot{\varepsilon }}^{e}+\dot{D}\circ \varepsilon^{e} \end{array}$$
where **σ** = **σ**(σ*_ij_*) is Cauchy stress tensor, *x_α_* is location of considered point, **D** = **D**(T) is a stiffness matrix. The total strain rate is decomposed by its constituents: elastic strain ε*^e^*, plastic strain ε^p^ and thermal strain ε^Th^.

## 4. Numerical Simulation

Numerical calculations for all case inclination of heat source are performed in Abaqus/Standard solver, extended with additional, self-made numerical subroutines. The discrete model is developed on the basis of the geometry used in experimental research. Varying with temperature material properties of X5CrNi18-10 austenitic steel are used in the material module of Abaqus. The modified Gauss model presented in Section 3 is included into the DFLUX subroutine. Laser beam power, welding speed, the inclination of laser beam are assumed on the basis of experimental research. 3D numerical model of analyzed case is developed for computer simulations of the process. The scheme of the analyzed domain is shown in Figure 7.

The dimensions of the numerical model are adopted in accordance with the experiment. The parameters of the heating source are assumed in accordance with the technological parameters of the welding process: beam power *Q* = 2100 W, beam inclination angle α = 20°, source travel speed *v* = 1.5 m/min. The influence of the source inclination on the weld pool shape and mechanical properties was analyzed. Three different source slope were adopted in the numerical simulation: α = 20°, 30° and 40°.

The Abaqus materials module takes into account changes of physical properties of welded element in the function of temperature and heat convection coefficient *k* = 100 W/m^2^. The thermomechanical properties of the welded T-joint presented in Figure 8 are assumed in numerical simulations for austenitic steel X5CrNi18-10 [32]. Chemical composition: 0.06 C, 17–19 Cr, 11–13 Ni, N < 0.11 [%].

The discrete model prepared in the software together with the finite element mesh (FEM) is present in Figure 9.

The densest finite element mesh is concentrated in the place of the greatest temperature gradients. The mesh changes linearly, increasing the size of the element with the distance from the weld line. About 370,000 finite elements are used in the developed model. The created finite element mesh allows us to minimize the duration of the numerical simulation while maintaining good quality simulation results. A simplification by applying the perfect contact between two joined surfaces is assumed in computer simulations. The analysis of thermomechanical phenomena is carried out using the “uncoupled” method. First, calculation of temperature field is carried out, followed by numerical analysis of mechanical phenomena. The results from the thermal analysis are the input data for the mechanical analysis. The mechanical boundary conditions were adopted in accordance with the boundary conditions contained in publication [34].

The temperature field and the shape of the FZ are numerically estimated. Figure 10 shows the temperature distribution in the general view for two different process duration times *t* = 3 s and *t* = 5 s. For the result presented in Figure 11, the beam inclination angle was α = 20°. For the same slope, Figure 11 shows the temperature field in ZY plane of the welded T-joint. In order to compare the results of numerical prediction with experimental tests, the boundary of melted zone (*T_L_* ≈ 1455 °C) is marked with a solid line. The numerically estimated shape of the weld is marked on the macroscopic picture of the joint. The comparison of FZ geometry with macroscopic view shows that a good representation of the weld shape is obtained. Good compliance of the calculation results with the experiment is obtained for the following source parameters: source penetration depth *d* = 2.8 mm, beam radius *r_t_* = 0.2 mm (for *z* = 0), beam radius *r_b_* = 0.08 mm (for *z* = *h*_1_), *h*_1_ = 4 mm.

The compatibility of results of numerical calculations and results of the experiment indicate the correctness of selected welding process parameters, as well as the correctness of the theoretical approach.

Deformations resulting from the heat load during welding are estimated on the basis of the determined temperature field for a welded T-joint. The analysis of mechanical properties of the joint was also carried out for three heat source inclinations.

On the basis of the simulations carried out in the Abaqus FEA program (ver.6.10) of mechanical phenomena, the stress state and the displacement field of the welded T-joint were determined. Figure 12a present temporary reduced stresses, while Figure 12b presents residual stresses.

The maximum value of these stresses is less than 220 MPa. The maximum concentration of stresses occurs in the welding line. The displacement area is numerically estimated for welded T-joint. Figure 13 shows the displacement field in the general view. Figure 13a shows averaged displacements, while Figure 13b shows vertical displacements (in accordance with the *Y*-axis). The largest displacements of 0.5 mm occur at the ends of edges of T-joint legs in final parts of welded joint.

Analysis of presented results of numerical prediction shows that the greatest values of displacements occur in the direction perpendicular to the weld line.

Figure 14 shows a macroscopic picture of welded joint, which marked thicknesses of joined flat bars, the axis of the weld is inclined at an angle α = 20° and dashed lines that indicate the shape of the system before welding. The comparison of plotted lines with the cross-sectional view of welded joint shows that both ends of the legs of T-joint and the vertical element of T-joint are displaced.

Figure 15 shows the displacements in the cross-section of analyzed T-joint. The numerically predicted directions of displacements confirm the results of the experiment (Figure 15). The arms of T-joint suffered the greatest displacements. The maximum value displacements in cross-section (the center of T-joint length) is 0.3 mm. Numerically estimated values of displacements well agree with experimental values. Figure 16 presents the results of numerical calculations of thermal phenomena for three different inclinations of the welding source.

Analyzing the results of the simulation of thermal phenomena presented in Figure 16, it can be concluded that the change in the beam inclination has a significant impact on the shape of the weld pool. Increasing the angle of inclination of the beam results increased penetration of the lower part T-joint which is most visible with the slope α = 40°. Changing the angle of inclination also reduces the melted zone on the opposite side of the vertical element of T-joint. Further increasing the angle of inclination may prevent full penetration. This contributes to the decrease of the quality of the welded joint.

The results of stress state and forecasting of the displacement of a welded T-joint are presented in the following figures. The results presented in these figures are determined for nine measurement lines, in accordance with the diagram shown in Figure 17.

Figure 18, Figure 19 and Figure 20 show the diagrams of residual stresses for the source inclinations α = 20°, 30° and 40°, respectively. Figure 18a, Figure 19a and Figure 20a show the results for lines (lines 1, 2, 3) perpendicular to the weld line of the joint, whereas Figure 18b, Figure 19b and Figure 20b describe the results for lines parallel to the welding lines (lines of the measurement 4–9).

The comparison of presented figures of reduced stresses, shows that the change in the inclination of the source has no significant effect on the change in the value of stresses in the joint. The difference between the values is approximately 5 MPa. Figure 21, Figure 22 and Figure 23 show the diagrams of displacement U_y_ for the source inclinations α = 20°, 30° and 40°, respectively. Figure 21a, Figure 22a and Figure 23a show the results for lines (lines 1, 2, 3) perpendicular to the weld line of the joint. Whereas Figure 21b, Figure 22b and Figure 23b describe the results for lines parallel to the welding lines (measuring lines 4–9).

The laser welded T-joint undergoes transverse and longitudinal deformation. The greatest displacements U_y_ occur in the direction perpendicular to the welding line. The differences in the obtained results are slight. Obtained simulation results are compared in Figure 24 in order to visualize the differences between displacement U_y_.

From the comparison of the simulation results shown in Figure 24, it can be noticed that the change of the beam inclination reduces the value of U_y_ displacements. For the analyzed model difference between slope α = 20° and 30° is about 0.03 mm.

## 5. Conclusions

Performing single-side laser beam welding of T-joints requires the selection of the correct heat source angle inclination to ensure a durable, good-quality joint. Determination of the influence of the inclination angle on the shape and size of melted zone by experimental research is very time-consuming and expensive. Abaqus FEA software is used for numerical modeling of the laser welding process of single-side welded T-joint. Development of a numerical model requires careful analysis of specific process conditions. The concept of using the basic welding process parameters from the experiment in simulations is crucial for verification of the model and allows for further correct prediction of the welding process parameters for various inclination angles. The paper presents the results of numerical prediction of the size and shape of melted zone, deformation and residual stresses of single-side welded T- joints for three different inclinations of the heat source. The end edges of the T-joint are subject to the greatest deformations. The maximum displacement values in the entire T-joint are 0.5 mm (Figure 13 and Figure 15). Results of calculated displacement are compared with displacements obtained in the experiment (Figure 15). As shown in these figures, a good consistency of the results of numerical calculations of deformation with the results of experimental tests is obtained.

From the analysis of the influence of changing the inclination of the heat source on the weld pool shape, deformation and stress state of the joint, the following practical conclusions can be reached:Changing the inclination of the heat source changes the shape and size of the melted zone. As the angle of inclination increases, the depth of fusion into the base of T-joint increases, and the width of the fusion zone decreases at the same time. A further increase in the angle of inclination of the source may lead to a reduction of the quality of the joint (see Figure 16).Changing the inclination of the heat source changes the deformation of the joint. Increasing the angle of inclination reduces the amount of deformation (Figure 24). For the analyzed joint, the differences are at the level of 0.03 mm.Changing the inclination of the heat source has no significant effect on the stress state (see Figure 18, Figure 19 and Figure 20). In general, increasing the angle of inclination of the heat source slightly increases residual stress. The predicted magnitudes and distributions of are determined only by numerical simulations, according to the model adopted in the Abaqus FEA software. Detailed analysis of the stress state in single-side welded T-joint requires experimental verification. Stress measurement in T-joints is complex, requires special equipment and will be the subject of further research.

## Figures and Tables

**Figure 1 materials-14-05303-f001:**
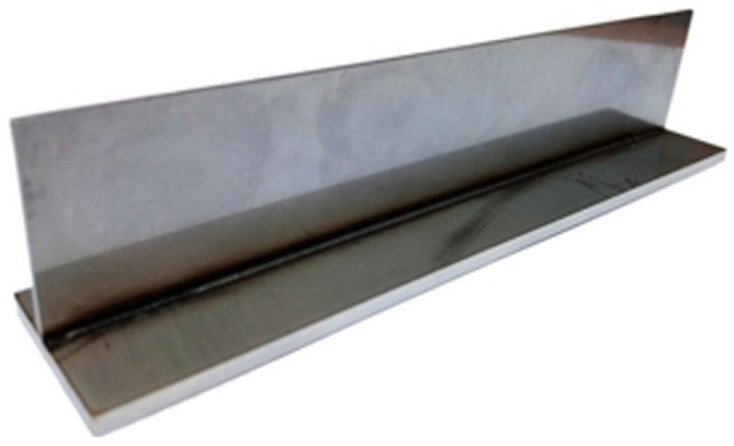
Single-side welded T-joint.

**Figure 2 materials-14-05303-f002:**
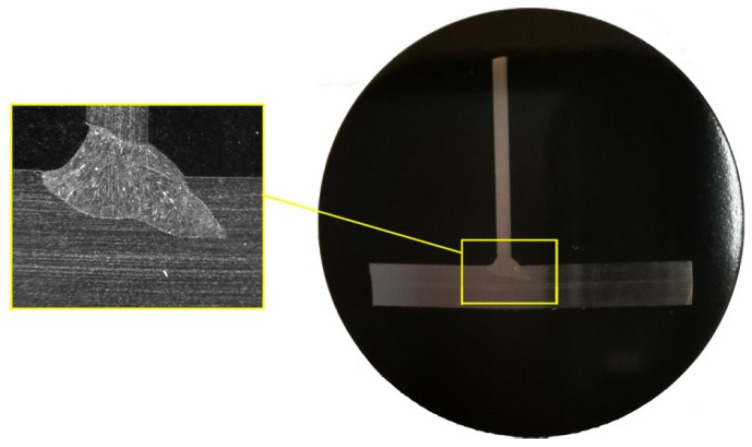
T-joint macrostructure in the cross-section.

**Figure 3 materials-14-05303-f003:**
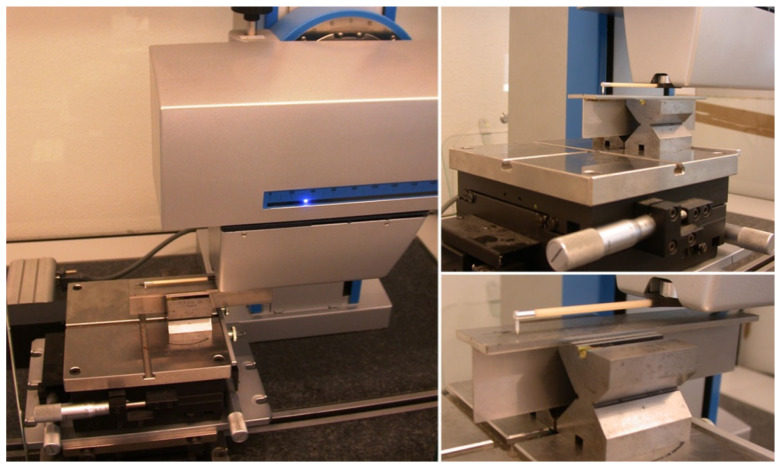
Displacement measuring system.

**Figure 4 materials-14-05303-f004:**
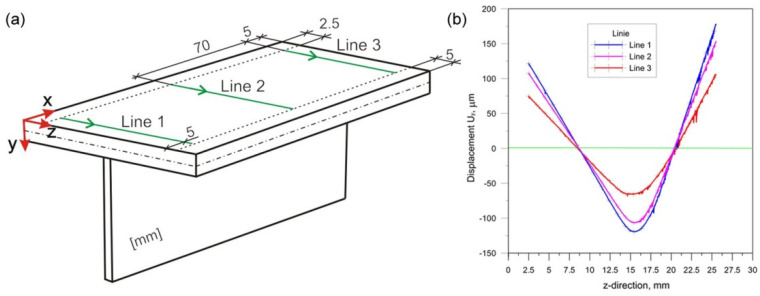
Scheme of lines of the measurement (**a**), measurement results (**b**).

**Figure 5 materials-14-05303-f005:**
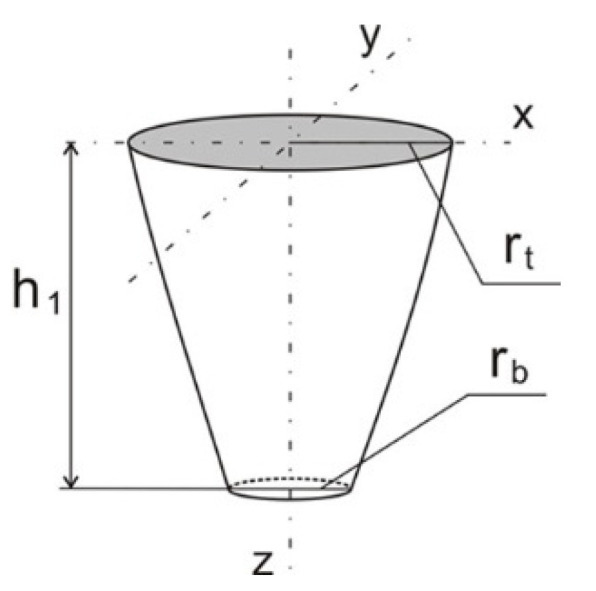
Scheme of laser beam heat flux distribution.

**Figure 6 materials-14-05303-f006:**
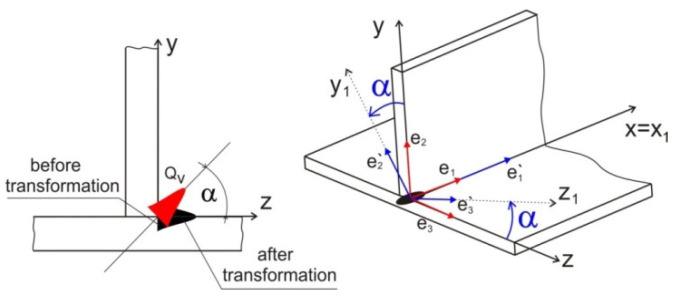
The schema of heat source coordinate transformation.

**Figure 7 materials-14-05303-f007:**
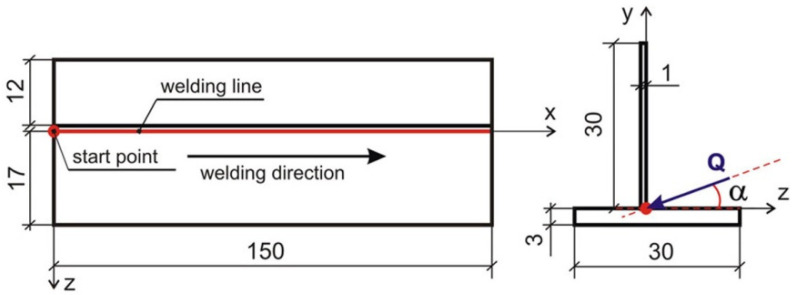
Scheme of analyzed domain.

**Figure 8 materials-14-05303-f008:**
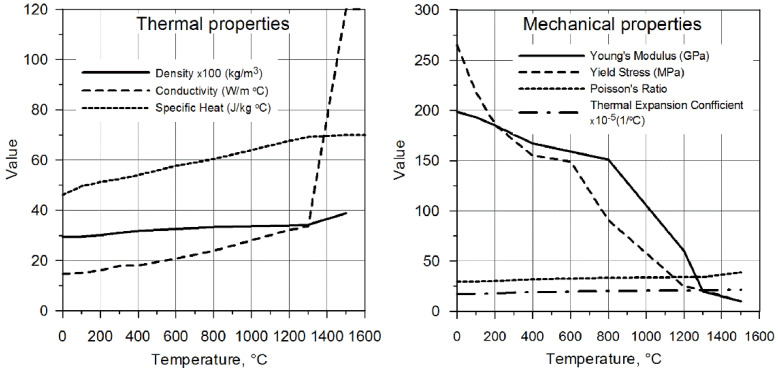
Thermomechanical properties assumed in calculations [32].

**Figure 9 materials-14-05303-f009:**
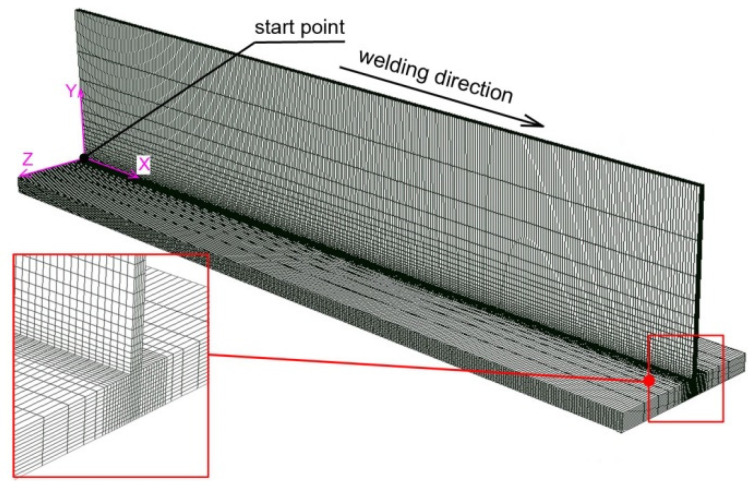
Scheme of analyzed domain. Discrete model with a FEM.

**Figure 10 materials-14-05303-f010:**
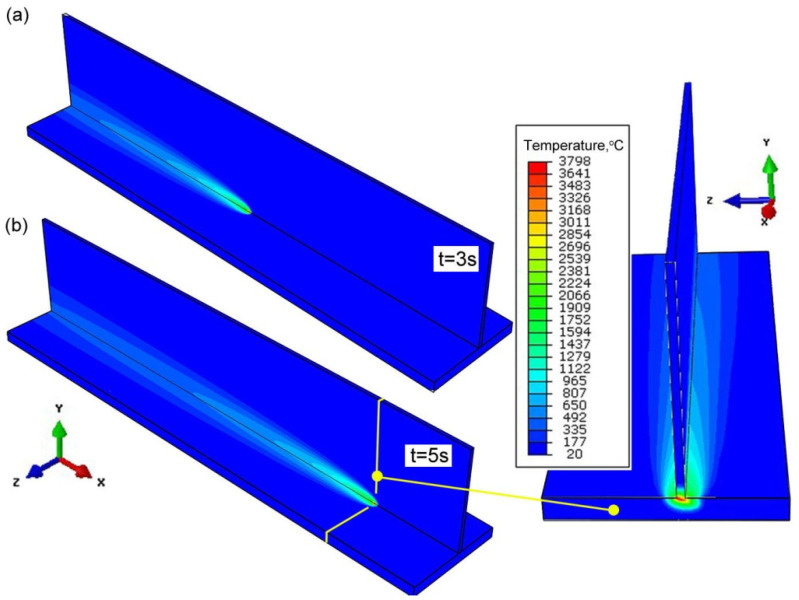
Temperature distribution in welded joint (**a**) *t* = 3 s, (**b**) *t* = 5 s (inclination angle α = 20°).

**Figure 11 materials-14-05303-f011:**
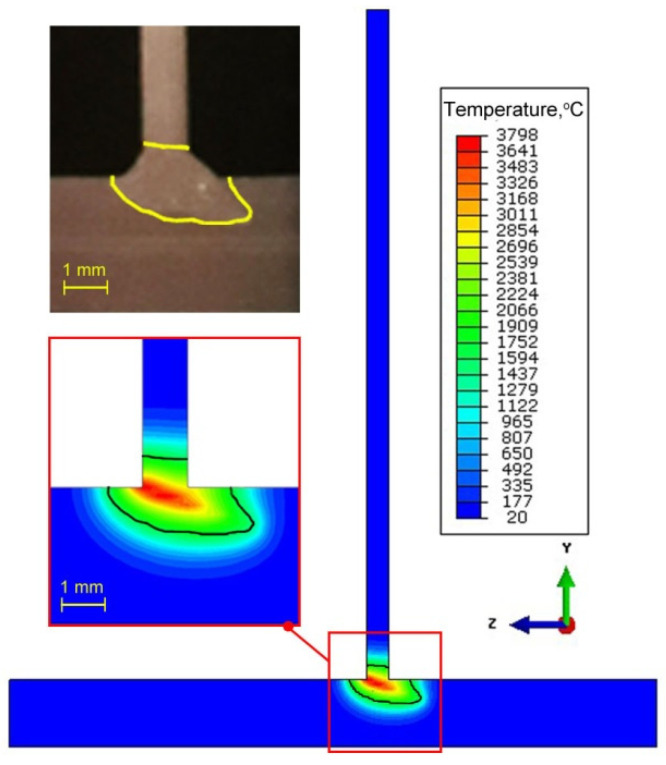
Temperature field in the joint cross-section and comparison of FZ with the experiment.

**Figure 12 materials-14-05303-f012:**
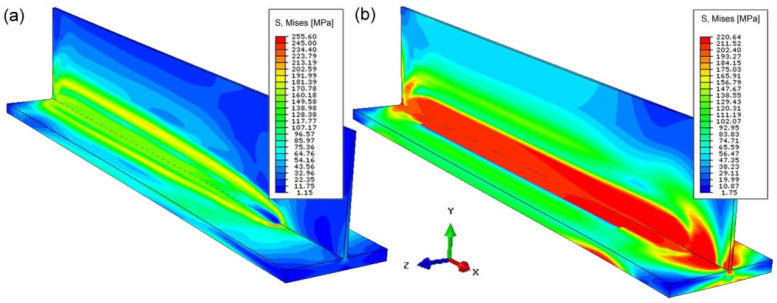
Residual temporary reduced stress (**a**) and reduced residual stress (**b**) of laser welded joint.

**Figure 13 materials-14-05303-f013:**
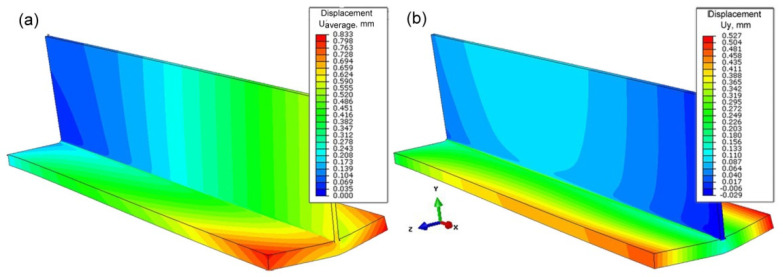
Displacement (**a**) average, (**b**) in *Y*-axis direction.

**Figure 14 materials-14-05303-f014:**
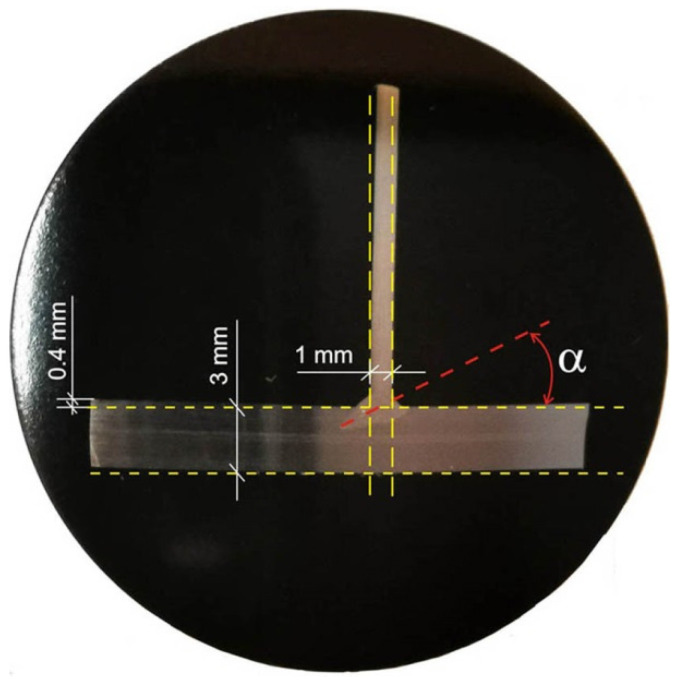
T-joint macrostructure in cross-section with shape marked before welding.

**Figure 15 materials-14-05303-f015:**
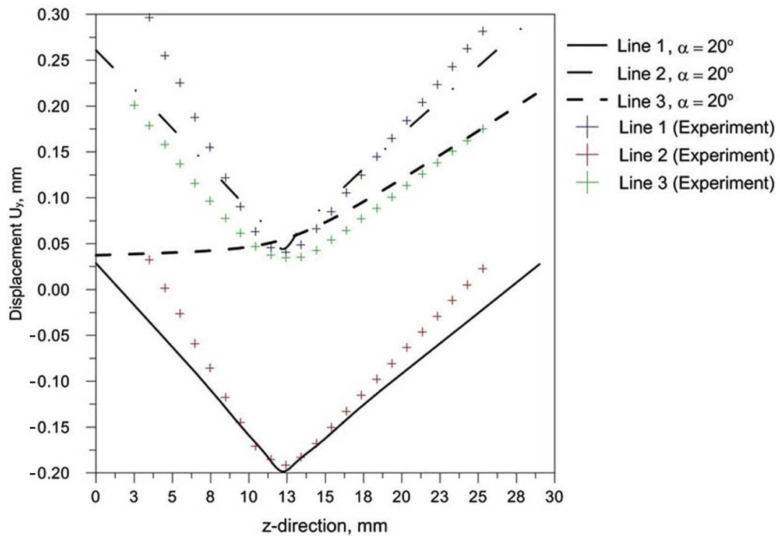
Compare numerical results with experience.

**Figure 16 materials-14-05303-f016:**
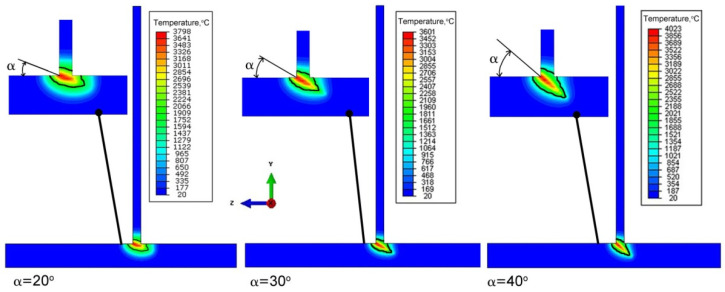
Comparison of the simulation results of thermal phenomena for different slopes of the heating source.

**Figure 17 materials-14-05303-f017:**
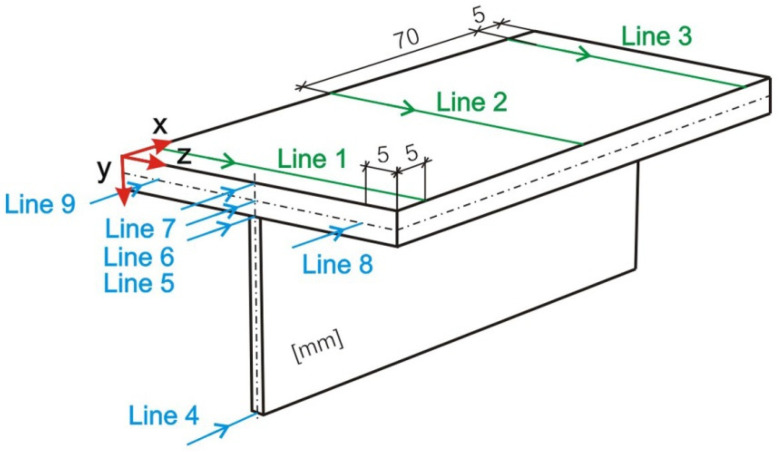
Scheme of measurement lines.

**Figure 18 materials-14-05303-f018:**
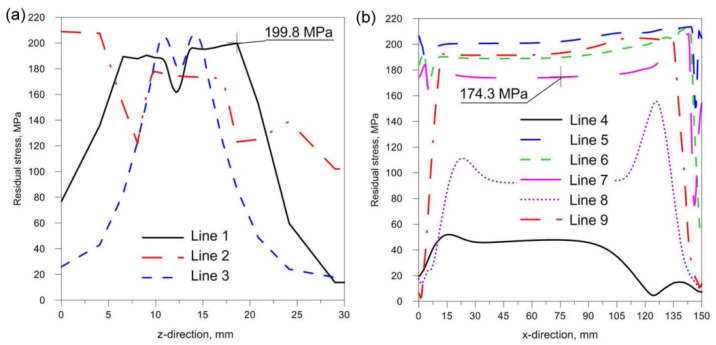
Reduced residual stress σ for slope α = 20°. (**a**) transverse direction to the welding line (**b**) along the welding line.

**Figure 19 materials-14-05303-f019:**
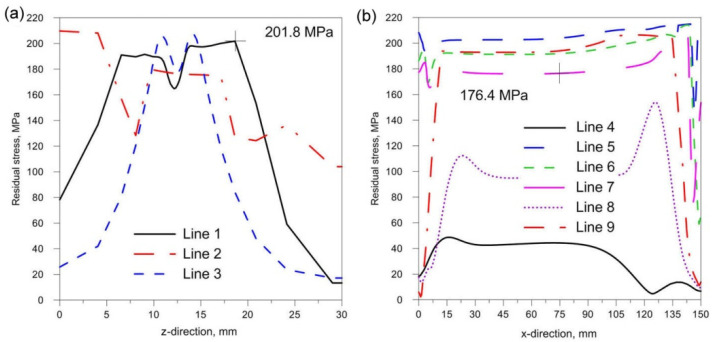
Reduced residual stress σ for slope α = 30°. (**a**) transverse direction to the welding line (**b**) along the welding line.

**Figure 20 materials-14-05303-f020:**
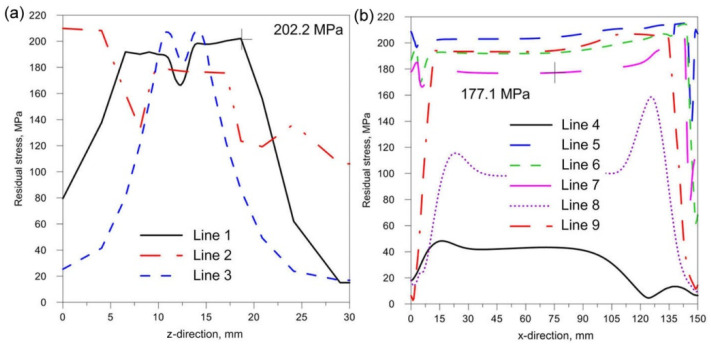
Reduced residual stress σ for slope α = 40°. (**a**) transverse direction to the welding line (**b**) along the welding line.

**Figure 21 materials-14-05303-f021:**
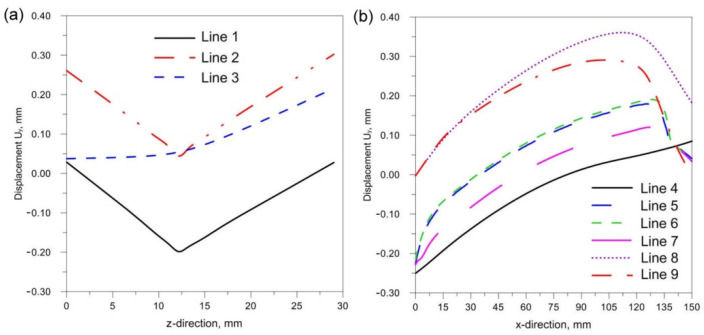
Numerically estimated displacement U_y_ in laser welded T-joint for slope α = 20°. (**a**) transverse direction to the welding line (**b**) along the welding line.

**Figure 22 materials-14-05303-f022:**
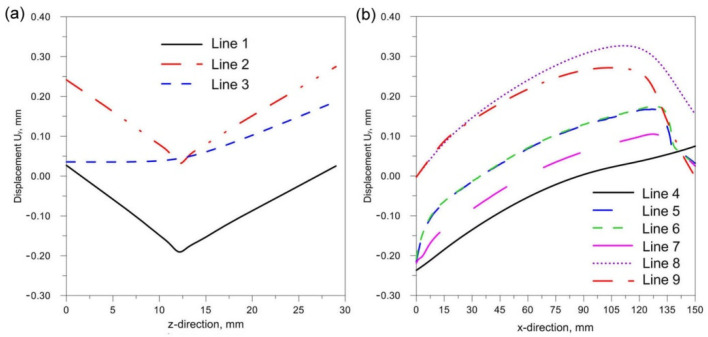
Numerically estimated displacement U_y_ in laser welded T-joint for slope α = 30°. (**a**) transverse direction to the welding line (**b**) along the welding line.

**Figure 23 materials-14-05303-f023:**
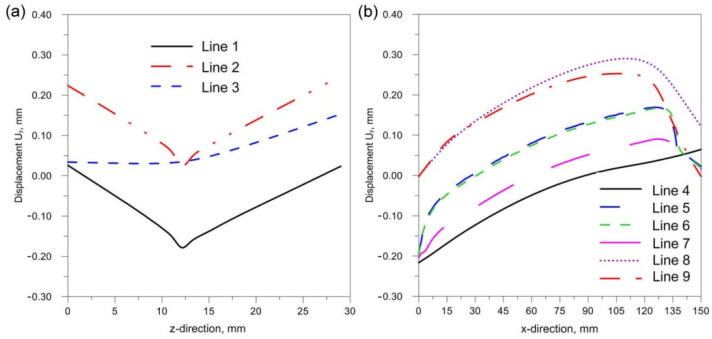
Numerically estimated displacement U_y_ in laser welded T-joint for slope α = 40°. (**a**) transverse direction to the welding line (**b**) along the welding line.

**Figure 24 materials-14-05303-f024:**
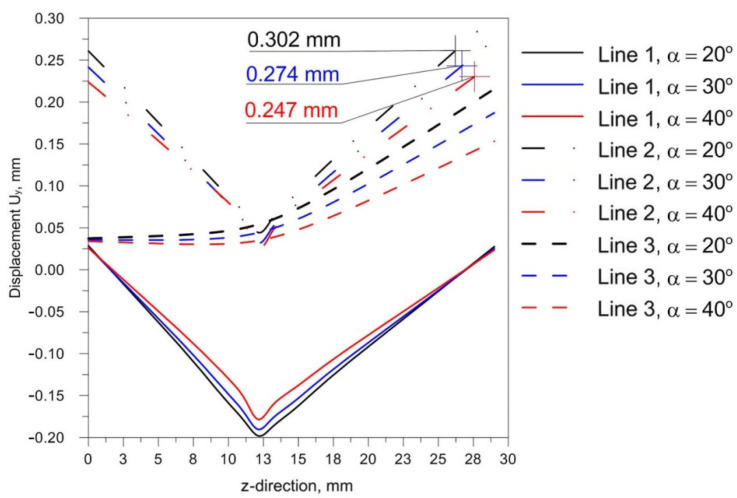
Comparison of numerically predicted displacement U_y_ for three different slopes α = 20°, 30° and 40°.

## Data Availability

Data is contained within the article.

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
