# Peer review of "The Influence of Welding Heat Source Inclination on the Melted Zone Shape, Deformations and Stress State of Laser Welded T-Joints"

_materials, 2021, doi:10.3390/ma14185303_

Round 1

Reviewer 1 Report

This work is important to guide the welding processing in engineering applications. And some comments are listed as:

  1. Paper title is not appropriate. The weld profile, residual stress and deformation of the weld joints are discussed in this paper, without comparative analysis of typical mechanical properties such as tensile property and impact property.
  2. FEA is important in simulating welding deformation and residual stress, and still is a core part of your work. But introduction of this content is not enough in Section 1 “Introduction”. Thus, some references you can add to enough the 5th paragraph of this Section, such as: Evolution Mechanism of Transient Strain and Residual Stress Distribution in Al 6061 Laser Welding, 2021, Crystals; Residual stress modelling in laser welding marine steel EH36 considering a thermodynamics-based solid phase transformation, 2018, International Journal of Mechanical Sciences. Magnetism aided mitigation of deformation and residual stress in dissimilar joint 316L with EH36, 2018, Journal of Materials Processing Technology.
  3. The influence of welding heat source inclination on the weld pool is discussed only based on FEA results. And how to verify the correctness of your simulated results? And Experiment results should be presented with different inclinations.
  4. Thermal distributions should be verified by experiment results (weld profile or thermal cycle curve).
  5. I suggest measured results of joint residual stress is added to this paper.
  6. Which kind of micro-phases in FZ and HAZ are consisted? Whether Solid-phase transformation are should be considered or not? Why?
  7. Coordinate systems and lines are not clear. In Figure 4b, the abscissa Z is inconsistent with Figure 4a. Line 7, 8, and 9 in Figure 15 was not marked earlier. Oddly, figure 17 contains line markings that contradict figure 4. Full text lines indicate confusion.
  8. There are problems with the units of many physical quantities. “...with the use of argon shielding gas with efficiency 16 l/min.”, “...is thermal conductivity (W/(m K)),...”, “... is convective coefficient (W/m2 °C),...”... The paragraph above figure 6 lacks a period. It is recommended that the author scrutinize the entire text.

Reviewer 2 Report

It is a properly written manuscript with detailed analysis and nice illustrations, however I am not fully convinced about the novelty of present paper, since basically a software application is introduced with scientific soundness.

The size of the samples were much smaller than the required dimensions in the EN ISO 15614 standard (350x150 or 300x125 mm). Probably the 30 mm seems a bit narrow. It can be assumed that this was due to material savings.  What is the authors' experience, can it have any detrimental effect on the simulation results? Can we expect a same temperature field if we use this reduced dimension compared to the size of the welding procedure test sample?

In the figures about the temperature fields (e.g. Figure 10.)  the highest temperature in the weld pool is 3798 °C. Is it a realistic value? Generally there are flow processes in the weld pool, so there is not a significant temperature gradient within it. I consider FEM simulaitons does not calculate with the flow processes. Therefore, during simulations generally liquidus temperature is set as maximum temperature in the scale for the visualization. In this case, the whole weld pool would belong to the same scale (and colour), which is benefical in terms of the analysis of the penetration and the shape.

Do you have information about the peak temperature of the weld pool in these stainless steel during laser welding?

Formalities:

Application of the proper degree sign is recommended in the whole text: "°"

Line 118 0.5mm/s => 0.5 mm/s (space is missing)

Line 159 modeling of fillet welded => modelling of fillet weld

Line 193 heating sourcs => heat source

Line 196 shape weld pool => shape of the weld pool or weld pool shape

Line 226 1455°C => 1455°C (space is missing)

In Figure 10. space is missing between the value and the unit (3s => 3 s; 5s => 5 s)

Reviewer 3 Report

  1. English should be significantly improved (rewritten).
  2. The title needs to be changed as it does not reflect the content of the manuscript.
  3. The manuscript contains separate results of experiment and numerical simulation, but there is no comparison and discussion of how they correlate with each other. Without this very important bond, the paper is not complete.
  4. In conclusions, the main results should be formulated accurately and concisely. In the presented edition, everything is very blurry and it is not clear what the essence is.

Reviewer 4 Report

  1. In Figure 11, in order to correlate the fusion zone between experimental and modeling, a scale bar needs to be provided.
  2. Please confirm the transition point of displacement of Y-direction is the same for each line during experiments compared to modeling.
  3. in the figure, please pride the fusion zone area and compare it with the modeling results. 
  4. It is recommended to measure the bead geometry such as leg length, throat, etc and compare it with the modeling results.

Round 2

Reviewer 3 Report

If the editors find it appropriate, the manuscript might be published, as no specific contraindications have been found.

Reviewer 4 Report

I appreciated the author's efforts on the revised manuscript.